# CNAViz: An interactive webtool for user-guided segmentation of tumor DNA sequencing data

Zubair Lalani[1☯], Gillian Chu[1☯], Silas Hsu[1], Shaw Kagawa[1], Michael Xiang[1], Simone Zaccaria[2,3]*, Mohammed El-Kebir[1,4]*

**1** Department of Computer Science, University of Illinois Urbana-Champaign, Urbana, Illinois, United States of America, **2** Computational Cancer Genomics Research Group, University College London Cancer Institute, London, United Kingdom, **3** Cancer Research UK Lung Cancer Centre of Excellence, University College London Cancer Institute, London, United Kingdom, **4** Cancer Center at Illinois, University of Illinois Urbana-Champaign, Urbana, Illinois, United States of America

☯ These authors contributed equally to this work.
* s.zaccaria@ucl.ac.uk (SZ); melkebir@illinois.edu (MEK)

**Data Availability Statement:** All data including source code are available on GitHub: https://github.com/elkebir-group/cnaviz.

## Abstract

Copy-number aberrations (CNAs) are genetic alterations that amplify or delete the number of copies of large genomic segments. Although they are ubiquitous in cancer and, thus, a critical area of current cancer research, CNA identification from DNA sequencing data is challenging because it requires partitioning of the genome into complex segments with the same copy-number states that may not be contiguous. Existing segmentation algorithms address these challenges either by leveraging the local information among neighboring genomic regions, or by globally grouping genomic regions that are affected by similar CNAs across the entire genome. However, both approaches have limitations: overclustering in the case of local segmentation, or the omission of clusters corresponding to focal CNAs in the case of global segmentation. Importantly, inaccurate segmentation will lead to inaccurate identification of CNAs. For this reason, most pan-cancer research studies rely on manual procedures of quality control and anomaly correction. To improve copy-number segmentation, we introduce CNAVɪz, a web-based tool that enables the user to simultaneously perform local and global segmentation, thus overcoming the limitations of each approach. Using simulated data, we demonstrate that by several metrics, CNAVɪz allows the user to obtain more accurate segmentation relative to existing local and global segmentation methods. Moreover, we analyze six bulk DNA sequencing samples from three breast cancer patients. By validating with parallel single-cell DNA sequencing data from the same samples, we show that by using CNAVɪz, our user was able to obtain more accurate segmentation and improved accuracy in downstream copy-number calling.

## Author summary

Copy-number aberrations (CNAs) are large genetic alterations that are pervasive in cancer and, therefore, have been the focus of several cancer research studies. Copy-number

**Funding:** G.C. was supported by the National Science Foundation Graduate Research Fellowship (1746047). M.E-K. was supported by the National Science Foundation (CCF-1850502 and CCF-2046488) as well as funding from the Cancer Center at Illinois. S.Z. was supported by the Rosetrees Trust grant reference M917. The funders had no role in study design, data collection and analysis, decision to publish, or preparation of the manuscript.

**Competing interests:** The authors have declared that no competing interests exist.

segmentation is a key step in the process of CNA identification, which consist in partitioning the genome into genomic segments with the same copy-number state. However, segmentation is challenging and the limitations of current segmentation algorithms lead to inaccuracies in the characterization of CNAs. In this paper, we introduce CNAVIZ, an interactive web-based tool that enables the user to edit segmentation solutions and overcome current limitations. We demonstrate the ability of a user to use CNAVIZ to improve segmentation solutions on both simulated and real data, analyzing six published bulk DNA sequencing samples from three breast cancer patients. Finally, we demonstrate that these improvements in segmentation solutions improve accuracy in downstream copy-number calling, enabling more accurate analyses of intra-tumor heterogeneity.

This is a *PLOS Computational Biology* Software paper.

## Introduction

Most tumor genomes are characterised by the accumulation of *copy-number aberrations* (CNAs), which are somatic genetic alterations that are pervasive across different cancer types with on average 44% of the genome being affected by CNAs in solid tumors [1–3]. While normal diploid cells typically have two distinct copies, or *alleles*, of every gene in autosomal chromosomes, each CNA can simultaneously alter the dosage of hundreds to thousands of genes by increasing (gain) or decreasing (loss) the number of copies of a large genomic segment, including chromosomal arms and whole chromosomes [4, 5]. Not only is the identification of CNAs a key step to understanding cancer evolution [1, 6–8], it may also inform the development of targeted therapies as CNAs can introduce novel vulnerabilities for cancer cells that can be exploited for drug design [9–11].

Currently, most cancer studies characterize CNAs in large cohorts of cancer patients by performing DNA sequencing of one or multiple tumor samples [1, 3, 7]. Specifically, these studies use two related signals observed for each contiguous genomic region, or *bin* [12] (Fig 1 (a)). First, the *read depth ratio (RDR)* is defined as the ratio between the observed and expected number of sequencing reads that align to a specific bin. As such, variations in the RDR values indicate changes in the total number of copies: an increase/decrease in the values of RDR between different bins indicates a higher/lower number of copies. Second, the *B-allele frequency (BAF)* is defined as the proportion of sequencing reads that belong to only one of the two alleles of the bin. A value of 0.5 is expected for normal heterozygous diploid bins since each allele is present in exactly one copy and half of the sequencing reads are expected to be sequenced from each allele. As such, a significant deviation from this expected value, called *allelic imbalance*, indicates the presence of CNAs that alter the proportion of copies between the two alleles. Thus, analyzing variations of RDR and BAF values across bins allows the identification of CNAs in cancer genomes. However, this is a challenging task for which several algorithms have been proposed.

The majority of current CNA calling algorithms are based on *local segmentation* approaches. The key idea is that CNAs generally affect large genomic segments that comprise multiple bins and, therefore, neighboring bins have an increased probability to be or not be affected by the same CNA. As such, algorithms for change-point detection have been proposed to identify CNA-based genomic segments by grouping neighboring bins that do not have higher than expected variations in RDRs and BAFs (Fig 1b). Examples of these algorithms for DNA sequencing data include ASCAT [13, 14], BIC-seq [15], Control-FREEC [16], TITAN [17] for

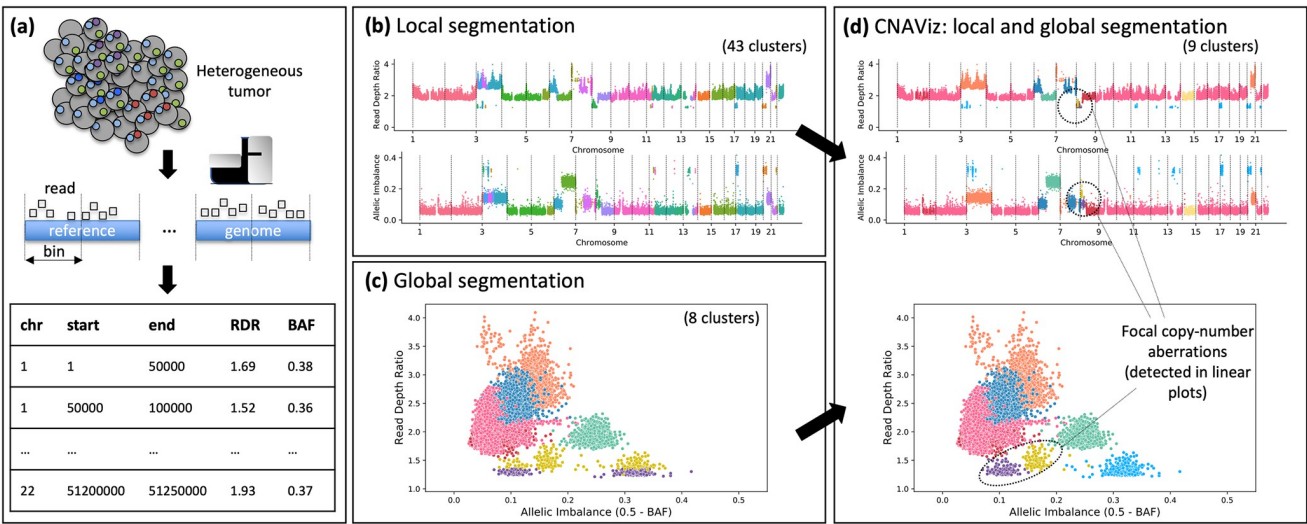

**Fig 1. CNAViz enables user-guided segmentation for improved copy-number calling.** (a) The genome of cancer cells (gray circles) is affected by CNAs (colored dots). DNA sequencing reads obtained from these cancer cells are aligned to a human reference genome, which is partitioned into bins (defined by the start and end position of the bin in a certain chromosome). For each bin, two signals are measured from DNA sequencing reads: the RDR, which is proportional to the total number of copies of the bin in the genome, and the BAF, which measures allelic imbalance. (b) Local segmentation algorithms combine neighboring bins with identical RDR (top plot) and BAF (bottom plot, where allelic imbalance is represented instead of BAF and is measured as $0.5 - BAF$) into segments. Differences across datasets might lead to overclustering. (c) Global segmentation algorithms cluster bins with similar RDR and BAF values across the entire genome, disregarding genomic location information, which may lead to spurious clusters and omit focal CNAs. (d) CNAViz allows the user to unify local and global segmentation approaches to obtain a more accurate segmentation.

bulk tumor samples. Additionally, methods such as HMMcopy [18] and Ginkgo [19] have been developed for single cell DNA sequencing data. The performance of local-segmentation algorithms can be substantially affected in different sequencing datasets by the presence of decreased or increased variance of RDR and BAF values between or within distinct genomic segments. While decreased variance is due to normal contamination, i.e. the presence of normal, non-cancerous cells in the sample [1, 13, 20], increased variance results from differences in sequencing technologies and platforms [21, 22].

To deal with the limitations of local segmentation, *global segmentation* approaches have been proposed, which leverage the presence of distinct genomic segments affected by similar CNAs. In fact, similar CNAs are frequent across the entire genome of the same tumor, resulting in bins from across the genome with similar RDR and BAF values. Thus, global-segmentation algorithms, such as FACETS [23] and CELLULOID [24], leverage these shared signals from different CNAs by clustering bins that share RDR and BAF values (Fig 1c). Moreover, the recent HATCHet [20] and CHISEL [22] algorithms have demonstrated that this global approach can be further extended to jointly leverage the signals even across multiple samples (or single cells) obtained from the same tumor, obtaining improved power to accurately identify CNAs even in the contexts of low tumor purity or CNAs that are only present in distinct subpopulations of cancer cells. However, this increased power afforded by global segmentation comes at the cost of a diminished ability to identify smaller or focal CNAs, as well as CNAs that are only present in few or single tumor samples, which are frequent in cancer [20]. Since local-segmentation algorithms generally have improved power for these smaller and focal CNAs by leveraging the local signals of neighboring genomic regions, there is thus a trade-off between local and global segmentation approaches.

Due to these and other challenges, copy-number analysis in practice often involves manual intervention and quality control. For instance, a recent pan-cancer study, PCAWG, covering

2,658 whole-genome sequenced human cancers, obtained consensus copy number calls from several algorithms through manual intervention to detect and correct anomalies [2]. Other examples include [7, 20, 24–26], where reported solutions were manually selected in order to balance the goodness of fit to data and proposed model complexity. Thus, while manual intervention in CNA calling is common practice, there is a lack of tools to facilitate this process, starting with enabling users to perform more accurate segmentation.

Here, we introduce CNAVIZ, a graphical, interactive, and web-based tool that enables users to perform manual segmentation of tumor DNA sequencing data for the identification of CNAs (Fig 1d). By providing an accessible and highly portable interactive platform to combine RDR and BAF values across both the entire genome and multiple samples while simultaneously revealing the presence of local genomic patterns, CNAVIZ represents a unifying approach that combines the advantages of local and global segmentation approaches. In particular, CNAVIZ is applicable to a wide range of novel and retrospective analyses, as it can be used to perform both segmentation *de novo* or to improve the segmentation performed by other existing segmentation methods. We have used simulated multi-sample tumor sequencing dataset generated by the published MASCoTE framework [20] to demonstrate the improved accuracy obtained with CNAVIZ relative to existing local and global segmentation methods. Moreover, we have applied CNAVIZ to previous bulk DNA sequencing data generated from 6 tumor samples obtained from 3 breast cancer patients [27]. Using these data, we have demonstrated that CNAVIZ enables the user to obtain a segmentation that results in CNA calls that are more concordant with parallel single-cell sequencing data of these samples, revealing the presence of CNAs for known breast cancer driver genes that would have been missed by current methods.

## Design and implementation

### Problem statement

In addition to sequencing a matched normal sample, one or more samples, quantified by $m > 0$, are sequenced from the tumor. DNA sequencing reads from these samples are then aligned to the reference genome, followed by partitioning of the genome into $n$ bins that may vary in size. We indicate the chromosome in which bin $i$ occurs by chr($i$), its start position on that chromosome by start($i$) and end position by end($i$). We extract two quantities from the alignment.

First, we obtain the *read depth ratio* RDR($p$, $i$) for each bin $i$ in each sample $p$, defined as the ratio between the normalized number of reads of bin $i$ in the sample $p$ vs. the number of reads in the matched normal sample. While RDRs are expected to be nearly constant in normal diploid cells, higher (lower) values of RDRs across the cancer genome allow the identification of corresponding gains (losses) due to CNAs. Second, by inspecting heterozygous germline single-nucleotide polymorphisms (SNPs), we obtain the *B-allele frequency* BAF($p$, $i$) for each bin $i$ in each sample $p$. As an example, if the BAF is observed to be 0.33 for a bin that is affected by a gain and has three copies (as indicated by the RDR), we can conclude that the genome contains two copies of one allele and one copy of the other; in contrast, a BAF of 0.0 would indicate that the genome contains three copies of only one allele.

An important preprocessing step in CNA callers is segmentation, which concerns the assignment of each bin $i$ to a segment or cluster, denoted by cluster($i$), based on its values RDR ($p$, 1), . . ., RDR($p$, $m$) and BAF($p$, 1), . . ., BAF($p$, $m$). Current methods perform this task in either a local or global fashion. While locality information of the bins is not utilized in global segmentation, it is used in local segmentation. The problems solved by both approaches can be summarized by the following two informal problem statements.

**Problem 1** (Local Segmentation). Given coordinates $< \text{chr}(i), \text{start}(i), \text{end}(i) >$, RDR and BAF values of $n$ bins in $m$ samples and integer $k > 0$, find an assignment $\sigma : [n] \to [k]$ of the $n$ bins into $k$ clusters with maximum likelihood such that the bins of each cluster $j \in [k]$ are contiguous in the reference genome.

**Problem 2** (Global Segmentation). Given RDR and BAF values of $n$ bins in $m$ samples and integer $k > 0$, find an assignment $\sigma : [n] \to [k]$ of the $n$ bins into $k$ clusters with maximum likelihood.

Local segmentation approaches are typically based on a Hidden Markov model or Circular Binary Segmentation, identifying change points via a parameter that controls the number $k$ of segments. On the other hand, global segmentation approaches view RDR and BAF values as a multi-variate mixture distribution, employing mixture models to identify the underlying $k$ composite distributions and clustering assignment. While global segmentation approaches are more robust to noise in lower coverage samples because they pool the signal across the genome, local segmentation approaches have the ability to detect small focal CNAs that global approaches may overlook.

Ideally, one would like to combine both approaches to overcome their respective limitations. Some methods, including FACETS [23] and CELLULOID [24], perform local segmentation followed by additional global clustering of the resulting local segments. Conversely, in Section B.4 in S1 Text, we describe a sequential Gaussian Mixture Model and Hidden Markov model approach, first performing global clustering into $k$ segments to obtain the $k$ composite distributions that best describe the mixture data followed by local segmentation. Unfortunately, all current automated approaches to segmentation still make mistakes that are easily identified via visual inspection. As mentioned in Introduction, current best practice consists of performing a parameter sweep and subsequently manually selecting a single solution among the results, often by inspecting each segmentation solution's goodness of fit with the data. Not only is this manual process time-consuming and labor-intensive, its inflexibility prevents the user from resolving inconsistencies in any one segmentation solution.

Rather than trying to improve segmentation and the downstream CNA calls by tweaking parameters which indirectly affect segmentation, we seek to enable the user to directly control segmentation via an interactive graphical user interface. Thus, CNAVIZ was designed as a web-based interface specifically to allow the user to directly cluster bins manually according to the dimensions of RDR and BAF, while also being informed by the genomic coordinates of these bins. The user can use CNAVIZ to either refine an existing segmentation or to perform *de novo* segmentation. To provide the user with direct control, our tool contains several critical features. First, the tool visualizes the RDR, BAF, and genomic coordinates of each bin. This task is achieved with a juxtaposition of three scatter plots, one for each combination of the relevant dimensions (RDR+BAF, RDR+coordinates, BAF+coordinates). Second, the tool allows the filtering and selection of bins along any of the three dimensions. Third, the user can manually cluster the bins by visual inspection, and edit each cluster as they see fit. Finally, the tool provides the user with cluster metrics that may help in optimizing cluster assignments. These additional features include the visualization of cluster centroids, driver genes by genomic position, assessments of cluster homogeneity and separation, and purity and ploidy estimation. Additional features and further details can be found in the appendix.

## CNAViz

This section details the functionality of CNAVIZ. Input and output defines the tool's inputs and outputs. Data exploration and design choices describes the ways in which CNAVIZ allows the user to visualize the data and interact with the clustering assignment, and provides

justification for the main elements of the CNAV<small>IZ</small> user interface. We describe the metrics used to evaluate each cluster in Cluster analytics, and discuss the automation of various cluster assignment tasks in Automation. Finally, we provide implementation details in Implementation details. We refer the reader to Section A in S1 Text for a complete list of CNAV<small>IZ</small>'s features.

**Input and output.** CNAV<small>IZ</small> takes two files as input and produces two output files. The main input is a tab-separated values (TSV) file containing the RDR and BAF values of bins across multiple samples. The first row specifies column headers, which must contain 'CHR', 'START', 'END', 'RD', 'BAF' and, optionally, 'CLUSTER'. The order in which these columns are specified does not matter. If the 'CLUSTER' is not provided, then we consider all the genomic bins to be un-clustered. That is, internally, we set cluster$(i) = -1$ for each bin $i$. As these files can be large (about 10 MB for $m = 3$ whole genome samples with $n = 53, 440$ bins of length 50 Kb), in order to process the data efficiently we require the rows to be ordered as follows: (1) All bins part of the same chromosome must be grouped together and sorted by genomic position. (2) Bins at the same genomic position, but from different samples are grouped together. (3) Every genomic bin should be present in every sample. Note that the TSV input file may contain additional columns, which will not be used, but will be included in exported files as discussed below. Furthermore, CNAV<small>IZ</small> includes a 'Demo' button that will load a published prostate cancer patient A12 [25]. We provide additional instruction on how to extract data in this format from alignment BAM files in our tutorial (https://github.com/elkebir-group/cnaviz). We have chosen a non-restrictive data input format, as most segmentation and copy number caller methods output these per-bin data. Therefore, the user has the option of providing a clustering of the bins output by any existing segmentation method. We provide conversion scripts and discuss how to obtain CNAV<small>IZ</small>'s input from ASCAT [14] [13] and HATCHet [20] in Section B in S1 Text.

The user may also optionally upload a list of driver genes to include in the visualization. The input data for driver genes must have the following columns: 'symbol' and 'Genome Location' where the latter column is of the format '{CHR}:{START}-{END}'. Note that this file is optional; the default list of driver genes corresponds to those genes in the COSMIC Cancer Gene Census (CGC) for which a genomic location was provided [28].

The user may export the current clustering. The exported file adheres to the same TSV format used for input and specifies the clustering. Bins $i$ that were erased, which we internally assign cluster cluster$(i) = -2$, will not be exported. The exported file will contain all columns, including any optional, user-provided columns that were previously imported. The user may also opt to download a text file containing a log of all clustering assignment operations that were performed.

**Data exploration and design choices.** As described previously, one of the primary goals is to support the clustering of genomic bins based on RDR and BAF while also being informed by the bins' genomic coordinates. CNAV<small>IZ</small>'s interface is composed of a hideable sidebar (Fig 2a–2d), a main view consisting of a main scatter plot (Fig 2f and 2i), and two linked scatter plots (Fig 2g and 2j). The main scatter plot compares the dimensions of RDR and *allelic imbalance*, equivalent to 0.5 − BAF. However, this main scatter plot lacks information about genomic coordinates. To address this challenge we place two scatter plots next to the main plot that plot the bins' genomic positions on the *x*-axis, and RDR and allelic imbalance on the y-axes respectively (Fig 2g). The total effect is that collectively, CNAV<small>IZ</small> visualizes the bivariate combinations of RDR and BAF, as well as the genomic coordinates of each bin in a sample. This juxtaposition of different scatter plots is an example of the well-known data visualization technique of using *multiple coordinated views* [29, 30]. This technique works well when no single view can perform all tasks and when juxtaposition can reveal new and insightful relationships

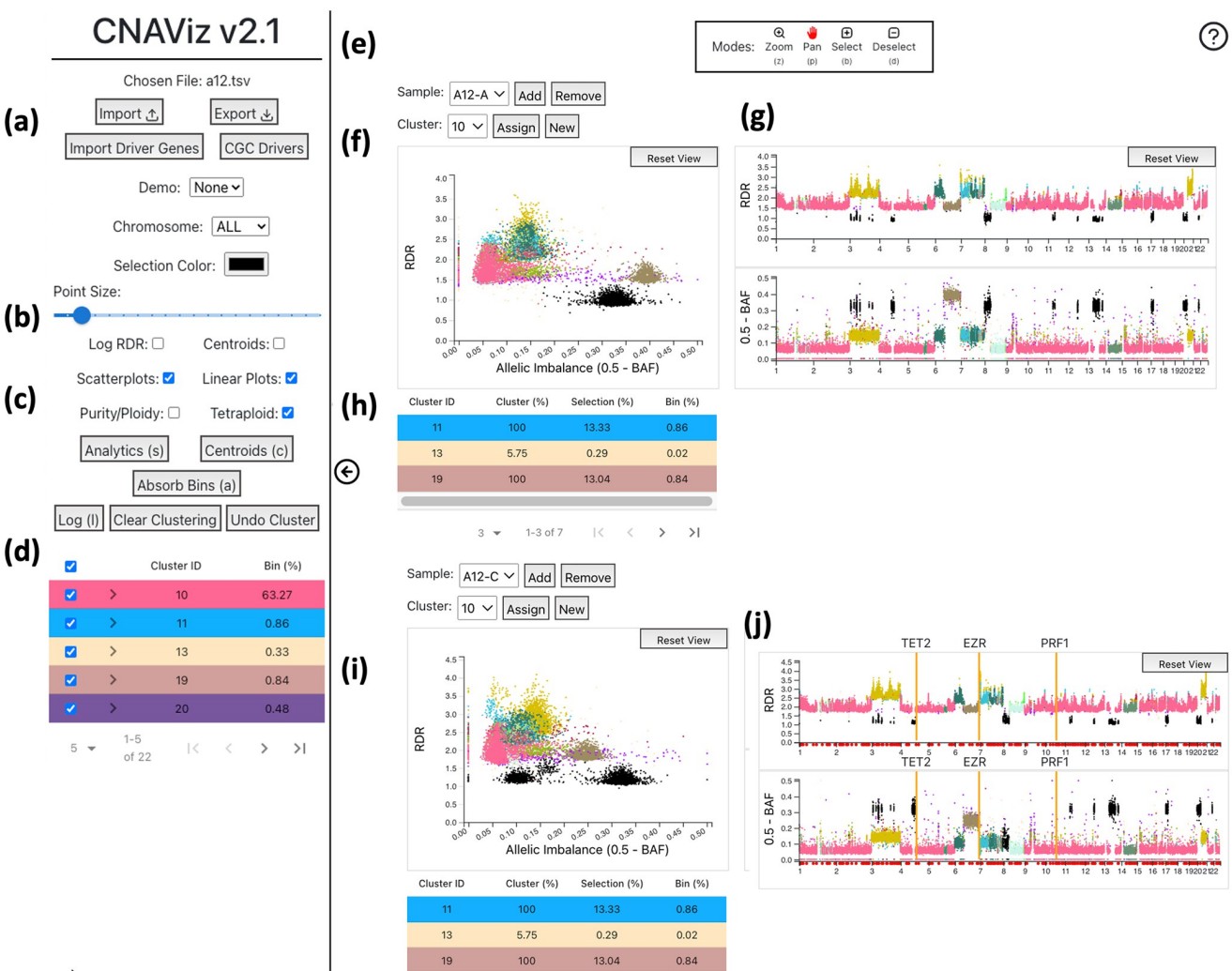

**Fig 2. CNAViz provides the user with a variety of options, modes, and plots to help the user create an effective segmentation.** (a) Buttons containing import/export options as well as a demo dataset, and allowing the user to import driver genes or use the existing Cancer Genome Census (COSMIC) driver genes. Also includes a drop-down menu for chromosome, the color of the selected bins (default is black), and point size of each bin. (b) Checkboxes controlling the 2D scatter and 1D linear plots. (c) Buttons which lead to pop-ups with analytics, automatic functions, and cluster assignment history. (d) A table summarizing all clusters assigned so far and the percentage of bins represented in each cluster. Also provides the user with the option to change the color for any cluster ID. (e) The toolbar at the top of the screen. The toolbar describing the different modes (Zoom, Pan, Select, Deselect), and their respective hotkeys, will float at the top center of the screen, and the help button is in the top right. (f) Scatter plot with RDR on the *y*-axis and allelic imbalance on the *x*-axis. When hovering over a point in the scatter plot, a tooltip appears with information about the corresponding bin including the genomic position, bin size, RDR, allelic imbalance, and cluster ID. In addition, the hovered bin's position on the linear plots is indicated with a black bar. (g) RDR and allelic imbalance linear plots with genomic position on the *x*-axis. (h) When points are selected, the color of the bins on all plots changes to a dark blue color. The cluster composition of the selected points is displayed under the plots with a table, where the row color matches the cluster color in the plots. (i) A second sample, where the selected bins are synced across the two samples and across the 2D scatter and 1D linear plots. (j) Driver genes are displayed as red dots along the x-axis of the linear plots. When a driver gene is clicked, it is locked in place and represented as an orange bar with the driver gene symbol above it. Hovering over one of the red dots allows the user to preview the driver gene (displayed as a green vertical bar).

from the data [29, 30]. In addition, all scatter plots color bins by their assigned cluster, and the user can add more triplets of scatter plots when they would like to visualize additional samples. Finally, to improve visibility, the user can adjust the point size via a slider in the sidebar.

Exploration of the data is critical for the user to perform segmentation efficiently. Two major themes inform our approach. First, our interface follows Ben Shneiderman's well-known visualization mantra for effective data exploration: overview first, zoom and filter, then

details-on-demand [31]. Second, our scatter plots are *linked* together; interactions in any one scatter plot affect all the other scatter plots across samples. Linking is prevalent in data exploration systems [32] and here it allows CNAVIZ's users to better understand how the data in the scatter plots relate to one another.

As the goal is to provide the user with a visualization of all the data, and moreover the use case is to resolve places where bins cluster one way in one sample and a second distinct way in another sample, CNAVIZ also allows the user to add and remove samples. Thus, the user can begin with an overview of genomic bins over all chromosomes and samples of interest. When the user becomes interested in a particular area, they can use the **pan** and **zoom tools**, which effectively function as filters. Keeping with our theme of linking, any change in the scale or range of an axis as a result of panning or zooming is reflected in all scatter plots relating to this sample. As a result, panning and zooming in one scatter plot, which can change which bins are in view, filters out the relevant bins in the other scatter plots for the same sample. In other words, we ensure all the scatter plots for a given sample always show the same set of bins.

An additional example of how CNAVIZ adheres to the principles of linking and details-on-demand, is that hovering over any bin will show details about that bin in a tooltip, and will emphasize that bin in all other scatter plots. In the two linear plots which show genomic position on the $y$-axis, this emphasis takes the form of a vertical black bar; alternative forms of emphasis, such as recoloring or increasing the point's border, were not visually salient enough. A critical feature for data exploration and editing is our **selection tool** and **deselect tool**. These tools allow the user to use the mouse to drag a bounding box (a "brush") to select and deselect bins inside any of the scatter plots. Selected bins are by default shaded black, which highly contrasts the default pastel colors assigned to each cluster. The user is also able to change this selection color in the sidebar. More importantly, the set of selected bins is highlighted across all scatter plots for all samples. This well-known general technique of *brushing and linking* [33] is essential for users to understand how points that are contiguous in one view are distributed and related in other views [30].

Once the user has selected the desired genomic bins, they can assign these selected bins to a new cluster. The "New" button will assign the next cluster ID available. Alternatively, users can choose a cluster from the drop down found above the scatter plot, and reassign the selected bins to the selected cluster ID by clicking "Assign Cluster". Cluster IDs -1 and -2 are reserved, each indicating a temporary "not clustered" state and a deleted state, respectively. As previously noted, those clusters in the $-2$ state will be excluded when the user exports the clustering assignment. The user may also clear all cluster assignments or undo their cluster assignments (or unassignments) with the respective buttons in the sidebar.

**Cluster analytics.** In order to allow users to see how well they are clustering the data, we introduce a 'Cluster Analytics' tab that shows the silhouette values of the clustering [34] as well as the distance between each pair of cluster's centroids. Specifically, given $m$ samples, we represent each bin $i$ as a vector

$$\mathbf{v}_i = [\mathrm{RDR}(1, i), \ldots, \mathrm{RDR}(m, i), \mathrm{BAF}(1, i), \ldots, \mathrm{BAF}(m, i)]^\top \qquad (1)$$

in $2m$-dimensional space, combining the $m$ RDR and the $m$ BAF values of the bin across all $m$ samples. This enables us to compute Euclidean distances between pairs of bins. To view analytics about the current clustering, the user can click the 'Analytics' button in the sidebar. A pop-up will appear that displays two bar plots (Fig 2f and 2g).

The first bar plot shows the approximated average silhouette coefficient for each cluster $j$. The silhouette value $s(i)$ of a bin $i$ is a value between $-1$ and 1, where a high value indicates that the bin is well matched to other bins assigned to the same cluster (homogeneity/cohesion)

and poorly matched to bins from other clusters (separation). The *silhouette coefficient s*(*j*) of a cluster *j* is the mean silhouette value of all bins *i* assigned to cluster *j*. Computing the exact silhouette coefficient of each cluster is time intensive, i.e. it requires $O(n^2)$ time where the number *n* of bins is around 50000 for real data. Therefore, we approximate the computation of the silhouette coefficient via downsampling of points. The goal is to obtain a clustering with silhouette coefficients near 1.

The second bar plot represents the average Euclidean distance between the points of two clusters, which enables the user to identify pairs of clusters that can be merged. From the drop down above the plot, the user chooses a specific cluster for which to compute distances to other clusters. Clusters that have a distance near 0 to the specified cluster are good candidates for merging. The goal is to obtain clusters that show good separation, and have large pairwise Euclidean distances. Finally, we provide the user the ability to visualize cluster centroids through a checkbox in the sidebar.

To further assess clustering, we allow the user to inspect clustering of bins containing driver genes. These driver genes are represented by dots along the *x*-axis of the linear plots. By default, we use the driver genes published in the COSMIC Cancer Gene Census, and restrict ourselves to those genes for which a genomic location was provided [28]. Each driver gene marker acts as a toggle button, where if toggled on, the driver gene's entire spanned genomic region is highlighted. When hovering over one of the markers, the highlighted region can be previewed (Fig 2j).

Finally, clustering can be assessed in terms of tumor purity and ploidy. The tumor purity is the proportion of tumor cells in a sample whereas the ploidy is the average number of copies. The estimation of these two quantities is a common but challenging step in all copy-number calling pipelines. We allow the user to vary values of tumor purity and ploidy for each sample, and subsequently estimate the integer copy-number states corresponding to the most common clonal copy-number states. This allows the user to pick better purity and ploidy values for the copy-number estimation process. We refer the reader to Fig A in S1 Text for a visual example.

**Automation.** Within the CNAViz user interface, we implement several automated tasks. First, the "Centroids" button, which can be found in the sidebar, enables the user to inspect cluster centroids locations and merge clusters according to centroid distance (Fig 3b). Specifically, the user may specify RDR and BAF thresholds for each sample. All pairs of clusters whose centroids' RDR and BAF values are located within the two user-specified thresholds for each sample, are flagged for merging. The user is prompted with a dialog box summarizing all clusters that will be merged if the action is taken. At this point, the user has the opportunity to abort the action, or to proceed with merging all the clusters together. To implement this functionality, we aggregate cluster pairs into connected components (e.g. if cluster 1 and 2 were identified to be merged, and cluster 2 and 3 were also identified to be merged, then 1, 2 and 3 form a connected component). For a single connected component set of clusters, the largest cluster is selected, and all other clusters' bins are reassigned to this cluster label.

While the previous functionality merged intact clusters, we provide additional functionality for splitting clusters. The "Absorb Bins" button, which can be found in the sidebar, allows the user to select "From" clusters, from which candidate bins will be drawn, and "To" clusters, to which candidate bins may be assigned (Fig 3c). For each bin *i* in a "From" cluster, we compute the RDR and BAF distance to its currently assigned cluster's centroid as well as to all "To" clusters' centroids. The bin is re-assigned if the distance to the nearest centroid meets the sample-specified specified BAF and RDR thresholds.

**Implementation details.** We implemented CNAViz in `React`. Each scatter plot was created using the `D3` (https://github.com/d3/d3) and `D3FC` (https://github.com/d3fc/d3fc) libraries. In order to give the user maximum control over the clustering, all bins from the input data

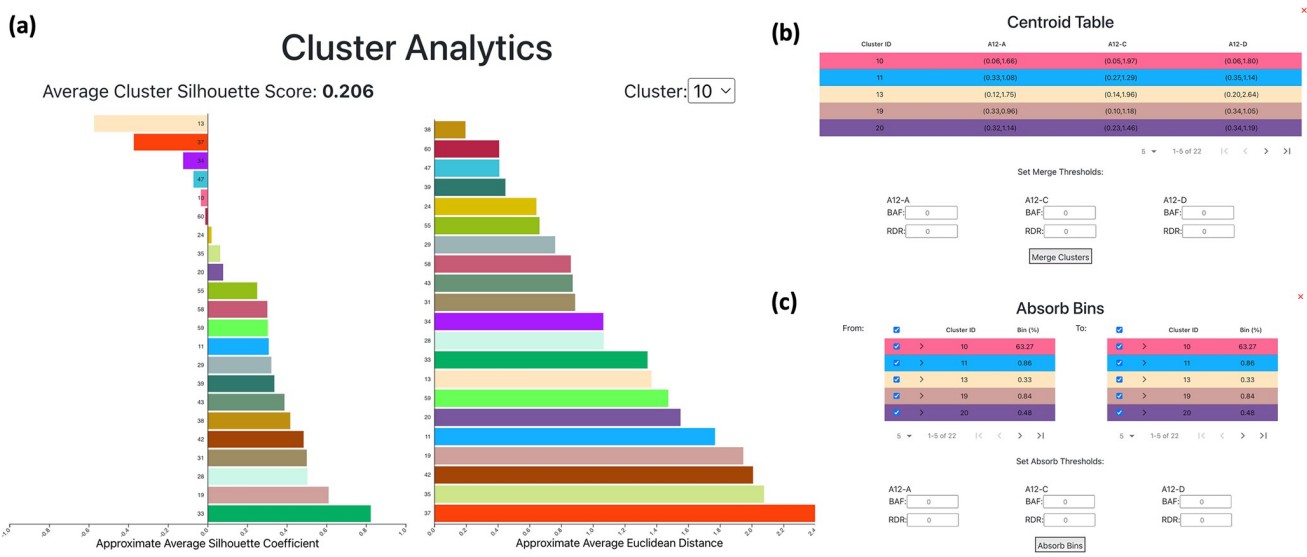

**Fig 3. CNAViz provides the user with a variety of analysis tools and automated functions to help generate an accurate segmentation.** (a) Average silhouette coefficient bar plot. Above the bar plot, the average of the silhouette scores for each cluster is displayed. Average Euclidean distance bar plot. Displays the average inter-cluster distance of each cluster to the cluster selected in the drop-down above the plot. (b) Centroid Table, illustrating each cluster, and the RDR and BAF values defining each cluster's centroid in each sample. In this pop-up, we also provide the user with the automated Merge function, which allows the user to set RDR and BAF thresholds per sample. Clusters whose centroids are closer than the user-defined thresholds will subsequently be merged. See Automation for further details. (c) The Absorb Bins pop-up allows the user to select "From" clusters and "To" clusters. All bins in the "From" clusters will be evaluated according to a user-defined threshold, and re-assigned to the closest legal "To" cluster. See Automation for further details.

are plotted without any merging or aggregation. We found that directly using SVG or drawing points using HTML Canvas does not scale to the number of bins that we have in our data ($n \approx$ 50,000 bins). In order to efficiently plot a large number of bins, we used `D3FC` wrapper methods for WebGL. WebGL takes advantage of the rendering speed of the GPU, which allows for the efficient rendering of large amounts of data points. Each plot in CNAVIz contains an SVG layer and WebGL layer to allow for both user interactivity and efficient rendering. On top of this architecture, we then accomplished tooltips with D3 quadtrees, and filtering with the `crossfilter` (https://github.com/crossfilter/crossfilter) library, which allows for filters along multiple dimensions to be added and removed with ease.

## Usage guidelines

We provide general guidelines on how users can apply CNAVIz in either *de novo* or refinement mode. Screencasts and detailed tutorials demonstrating the application of these guidelines on real and simulated data are publicly available and can be found at https://github.com/elkebir-group/cnaviz.

**Using CNAViz to perform *De Novo* segmentation.** We begin by providing guidelines for users to perform *de novo* segmentation using CNAVIz. We recommend displaying all samples in order to evaluate bins across samples concurrently. Moreover, we recommend using the scatter plot to quickly identify potential clusters that share similar RDR and BAF values across samples at a glance. However, the use of linear plots is essential to refine this clustering, especially in the presence of large number of clusters or clusters corresponding to small CNAs. Thus, both the scatter and linear plots should be used in the process of selecting relevant bins in the following three steps.

First, the user should select bins that are well separated on the scatter plot of a single sample. The user should then inspect whether these selected bins are also grouped together in other samples. In particular, selected bins that vary in one sample should be excluded from the current selection, and are good candidates for a new cluster. Second, the user should also use the linear plots to inspect whether these selected bins share RDR and BAF values across the genome. The linear plots are especially helpful to leverage the intuition that CNAs tend to occur in contiguous segments of the genome. Third, selected bins which share RDR and BAF values across samples can be made into a new cluster. This process should be repeated until each bin has been assigned to a cluster. When all bins have been clustered, the user can then proceed with the following steps to check an existing clustering.

**Using CNAViz to refine an existing segmentation.** We now provide a few guidelines with which to evaluate and improve upon an existing clustering. The user should begin by displaying all samples. As a first step, the user should toggle the plots to show only the bins in one chromosome. This can be achieved using either the sidebar's chromosome menu, or via the zoom selection. The following steps should then be repeated for each chromosome.

First, if a pair of clusters share both RDR and BAF values across all samples, these clusters should be merged. The user may find the following subroutine for merging clusters helpful. (1) Note the cluster IDs in question. (2) Use the cluster check boxes in the left toolbar to visualize only the bins in these clusters. (3) Use the 'Reset View' button to ensure all cluster bins are visualized. (4) Select all bins and either assign them to an existing cluster or create a new cluster as appropriate. (5) Repeat this process as necessary.

It should be noted that we provide the user with automated functionality to perform a related task. In particular, users can provide a sample-specific RDR and BAF threshold value, and automatically merge any cluster pairs whose centroids are closer than this threshold. For further details, please refer to Automation.

Second, if a single cluster contains different RDR and BAF values, this cluster should be split into at least two clusters. We suggest the following procedure for splitting clusters. (1) Note the cluster ID in question, and the approximate corresponding range of RDR and BAF for each new cluster. (2) Use the cluster check boxes in the left toolbar to visualize only the bins in this cluster. (3) Use the 'Reset View' button to ensure all cluster bins are visualized. (4) Select the bins that should be separated, and create a new cluster. (5) Repeat this process as necessary so that each cluster has distinct RDR and BAF values.

For this procedure, we also provide the user with automated functionality to make this operation more efficient. The user can specify clusters "From" which bins should be evaluated. For each such bin, the distance to a set of user-specified candidate centroids is calculated, and the minimum distance centroid is identified. If the distance between this bin and the minimum distance centroid is within the user-specified threshold in every sample, the bin is reassigned. For further details, we refer the reader to Automation.

Third, in an input clustering with several clusters which each have very few bins, it is often desirable to lessen the number of clusters by absorbing small clusters into larger ones. This is particularly relevant after inspecting and splitting each cluster, which results in the creation of several small clusters. The user should first verify that the largest clusters that incorporate the majority of bins are appropriately clustered—that is, each cluster's bins share a RDR and a BAF value that is distinct from all other bins. Next, given a small spurious cluster we suggest using the 'Analytics Tab' to identify a candidate largest cluster for merging. Finally, we recommend the user to iterate through these three steps until convergence. This last described procedure can be accomplished using a combination of the existing automated tools, so we do not provide additional automation here.

## Results

We used published simulated datasets [20] generated from multi-sample DNA sequencing tumor samples to demonstrate how CNAVIZ enables users to improve upon existing segmentation algorithms in Validation of CNAVIZ using simulations. Moreover, in Application of CNAVIZ to real data we demonstrate on a dataset of 6 tumor samples from 2 breast cancer patients that by using the novel features of CNAVIZ, we were able to accurately reveal CNAs affecting important cancer genes, which were previously missed by existing segmentation algorithms.

### Validation of CNAViz using simulations

**Experimental setup.** To demonstrate what CNAVIZ enables users to do, we used previously published data simulated with MASCoTE [20] for which ground truth is available and can be used for assessing segmentation performance. We considered the published dataset `n2_s4669/k4_01090_02008_00506035_00504055` with $m = 4$ bulk DNA sequencing samples comprising of 2 tumor clones.

To assess how CNAVIZ enables users to perform accurate *de novo* segmentation as well as to assess improvement upon segmentations produced by existing methods, we performed three different experiments. We first used CNAVIZ in *de novo* mode by providing non-segmented data as input and performing manual clustering in the user interface. Second, our user leveraged CNAVIZ to perform manual refinement of a segmentation solution generated by HATCHet, which performs global segmentation [20]. Third, we input a segmentation solution generated by ASCAT, which performs local segmentation [13, 14], and used CNAVIZ's user interface to perform refinement. We ran ASCAT in single-sample mode (aspcf) and provided it with ground-truth purity and ploidy values. We reconciled the sample-specific segmentation into a single sample-agnostic segmentation solution by retaining all breakpoints. We refer the reader to https://github.com/elkebir-group/cnaviz for screencasts describing the specific steps taken for this simulation instance. These follow the general guidelines described in Usage guidelines.

**Results.** We evaluated the different clustering solutions using three performance metrics. These include the Adjusted Rand Index (ARI) [35], the V-measure [36] and the silhouette score [34]. The ARI equals 0 when points are assigned to clusters randomly, and equals 1 when the inferred and ground-truth clustering solutions are the same. Likewise, the V-measure ranges from 0 (poor clustering) to 1 (matching ground-truth) [36]. We refer to Cluster analytics for further details on interpreting the silhouette score.

We assessed the performance of five different segmentation solutions produced by (i) CNA-VIZ, (ii) HATCHet, (iii) HATCHet + CNAVIZ, (iv) ASCAT, (v) ASCAT + CNAVIZ (Fig 4a). Notably, the segmentation produced manually clustering using CNAVIZ's *de novo* mode achieved the best overall clustering performance in terms of ARI and V-Measure (0.99553 and 0.97048, respectively). Given an existing solution, manual refinement using CNAVIZ also produced consistent improvements when compared to the original solution. Specifically, using CNAVIZ to perform manual refinement produced the greatest improvement in terms of both ARI and V-measure (0.07376 to 0.99509 for ARI, and 0.21984 to 0.96804 for V-measure) when applied to the ASCAT solution. We also see modest improvements in these metrics for HATCHet.

Next, we present two specific examples of typical errors made in existing methods that manual refinement using CNAVIZ is able to fix (Fig 4). First, CNAVIZ enables the user to improve the HATCHet solution by splitting a cluster. By visualizing the HATCHet solution using CNAVIZ's integrated scatter and linear plots, we can observe an orange cluster

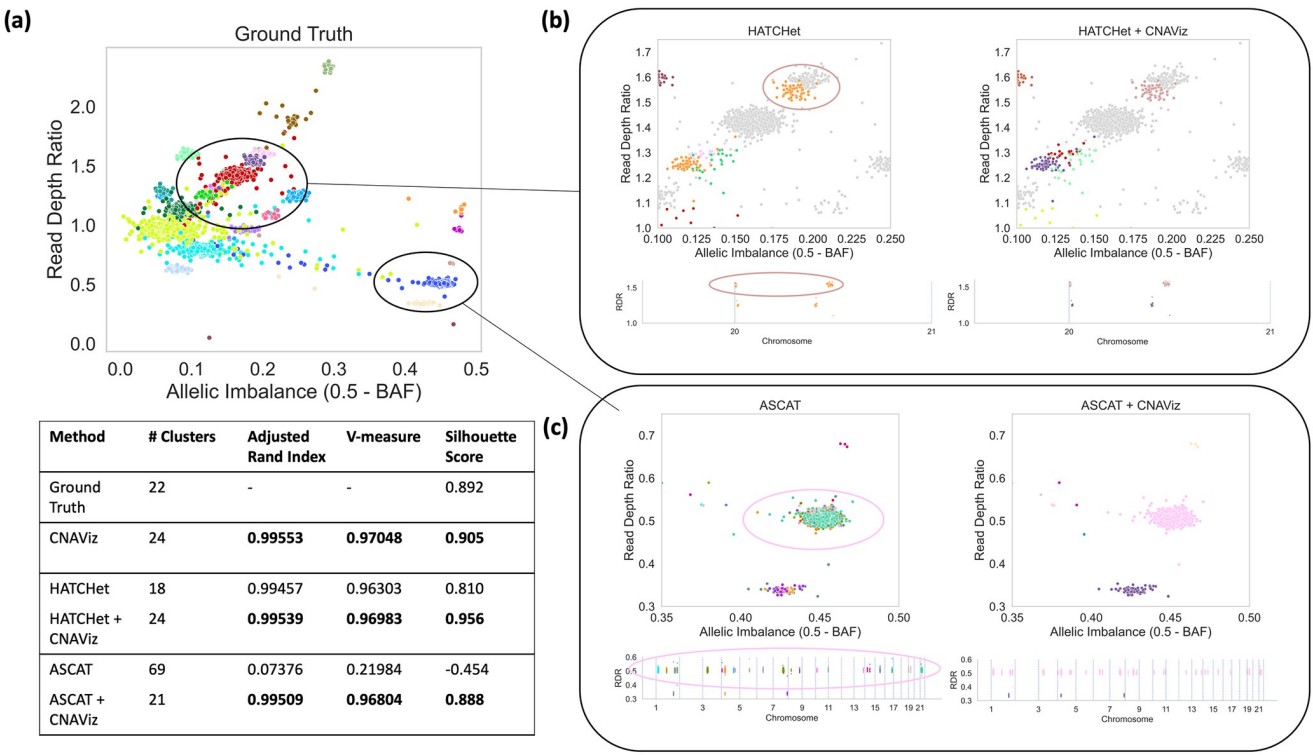

**Fig 4. By using CNAViz, users are able to produce more accurate segmentation solutions on simulated data in both *de novo* mode as well as when refining a given segmentation.** (a) A two-dimensional plot of RDR (*y*-axis) and allelic imbalance (*x*-axis, measured as $0.5 - \mathrm{BAF}$) of 50 Kb genomic bins (points). Colors represent the ground-truth segments/clusters. Table shows performance metrics for each method. (b) Comparison of HATCHet's global segmentation solution before (left plots) and after user refinement (HATCHet + CNAVɪᴢ, right plots). (c) Comparison of ASCAT's local segmentation solution before (left plots) and after user refinement (HATCHet + CNAVɪᴢ, right plots). In each plot of (b) of (c) respectively, the same genomic bins are displayed, but colored according to each method's inferred segmentation.

containing bins that separate into two distinct genomic segments along the genome (Fig 4b). Therefore, we split the orange cluster into two separate clusters (Fig 4b), matching ground truth (Fig 4a). Second, CNAVɪᴢ enables the user to combine distinct segments from across the genome into a single cluster. As a local segmentation method, ASCAT overclusters a single ground-truth cluster into 22 separate segments. ASCAT produces this clustering because the bins occur non-contiguously (Fig 4c). With CNAVɪᴢ's interactive scatter plot, we are able to both identify and reassign the cluster of bins (Fig 4c), producing a cluster that matches ground truth (Fig 4a).

For runtime estimates, we refer the reader to the accompanying recorded videos of manually editing the simulated sample s4669. Our first year graduate student with previous CNA calling experience completed segmentation in *de novo* mode in approximately 15 minutes, given a HATCHet initial clustering it took 20 minutes, and given an ASCAT initial clustering it took 1 hour.

## Application of CNAViz to real data

To investigate the impact of what CNAVɪᴢ's novel features enable the user to do on real data, we used CNAVɪᴢ to manually refine DNA sequenced from six tumor samples across three breast cancer patients (P5, P6, P10) analyzed in the previous study of [27]. In addition to standard bulk DNA sequencing of each tumor sample, the authors also performed matched high-resolution single-cell sequencing of every sample. As such, we can use these single-cell data to

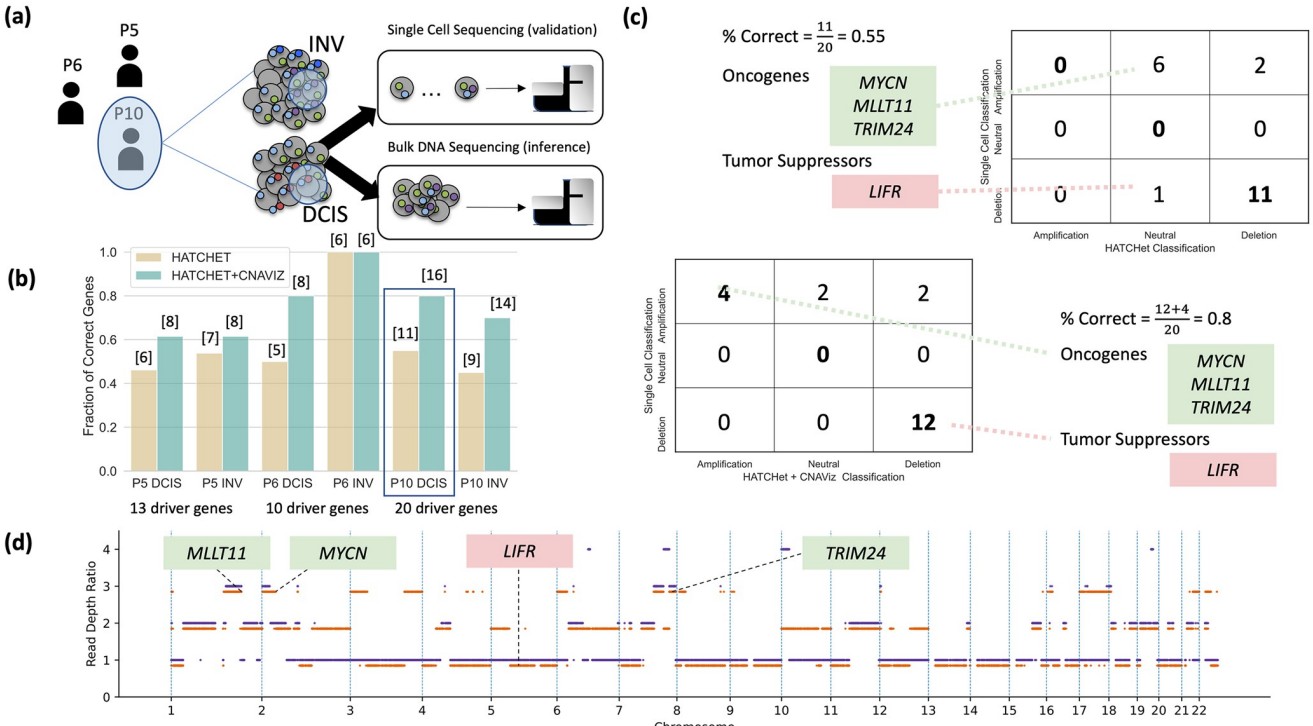

**Fig 5. Manual editing using CNAViz results in more accurate identification of CNA status of breast cancer driver genes compared to an existing segmentation algorithm.** The DNA sequencing data of two tumor samples (DCIS and INV) obtained from each of three breast cancer patients (P5, P6, and P10) analyzed by [27]. (b) The number of correctly identified CNAs for breast cancer driver genes (*y*-axis) is reported across all samples of the three patients when using either the existing segmentation algorithm HATCHet (yellow) or after manual refinement of the HATCHet results with CNAVɪz (green). The number of correct driver genes is listed above each bar. (c) The number of breast-cancer driver genes with different types of CNAs inferred by either HATCHet (columns in top table) or HATCHet + CNAVɪz (columns in bottom table) is compared with the high-resolution CNAs measured by the matched classification in single-cell sequencing data (rows in both tables). (d) The CNAs (*y*-axis) inferred by HATCHet + CNAVɪz for two distinct sub-populations of cancer cells identified in Patient 10 are shown in orange and purple, with 0.15 separation for visual clarity.

validate the CNAs inferred from the bulk sequencing data. Specifically, we plan to assess whether performing segmentation using CNAVɪz produces downstream CNA calls that better match the single-cell data compared to using an existing segmentation method (Fig 5a).

We processed the raw sequencing reads using the same pipeline reported in [27]. After downloading the DNA sequencing data from the Sequence Read Archive (accession numbers SRP114962 and SRP116771), we aligned the reads to the human reference genome (hg19) using BWA [37]. Then, the aligned sequencing reads were provided as input to HATCHet [20]. Similar to other methods for copy number calling, HATCHet first performs segmentation before outputting copy number calls. Due to its modular design, it is possible to provide HATCHet with a custom segmentation. We created two sets of CNA calls for each patient. One set was obtained by running HATCHet end-to-end with its built-in global segmentation (denoted as 'HATCHet'). We extracted HATCHet's global segmentation and manually refined it using CNAVɪz (following the guidelines in Usage guidelines). This enabled us to obtain a second set of CNA calls from HATCHet using the refined segmentation (denoted as 'HATCHet + CNAVɪz'). Although runtime estimates vary by user, it took our first year graduate student with previous CNA calling experience approximately 30 minutes to use CNAVɪz to manually edit each sample.

For each patient, [27] reported a small number of relevant breast cancer driver genes (ranging from 13 to 20). Using the single-cell CNA calls reported by the authors, we classified the

driver genes of each patient as either unaffected, deleted, or amplified due to CNAs. We designated a driver gene as correctly classified if the CNA state inferred from bulk data matched the single-cell CNA state. We found that manually refining the HATCHet clustering using CNAVIZ (HATCHet + CNAVIZ) classified a total of 60/86 genes (70%) compared to 44/86 genes (51%) correctly classified by HATCHet alone (Fig 5b). In particular, for sample P10 DCIS (ductal carcinoma *in situ*) using HATCHet + CNAVIZ enabled the user to produce a manual clustering with 16 genes correctly inferred compared to 15 genes correctly inferred by HATCHet without manual refinement. Further inspection reveals that HATCHet alone identified no amplified genes, and instead identifies 7 driver genes as neutral and 13 driver genes as deletions (Fig 5c and 5d). By contrast, by having a user manually refine a HATCHet clustering solution using CNAVIZ (HATCHet + CNAVIZ), we identified 4 amplifications among driver genes, matching the ground-truth single-cell data. Among these, three are known oncogenes: *TRIM24* [38], *MYCN* [39] and *MLLT11* (also known as *AF1q*) [40]. Generally, we expect oncogenes to be amplified within tumor cells, as these mutations prove beneficial to tumor cells. Thus, the literature provides further evidence corroborating the manually refined HATCHet + CNAVIZ's classification of these genes. Another difference between both approaches is the classification of the driver gene *LIFR*, which is a known tumor suppressor gene [41]. While HATCHet classified this gene as unaffected by CNAs, the manually refined HATCHet + CNAVIZ solution classified the gene as affected by a deletion. This matches the expected behavior for tumor suppressor genes, which are frequently affected by deletions.

In summary, significant improvements in the accuracy of downstream copy-number analyses are possible with more accurate upstream segmentation. Here, we have illustrated improvements in the use case of driver gene classification, made possible by using CNAVIZ to manually refine the segmentation prior to copy number calling.

## Availability and future directions

Here, we introduced CNAVIZ, a web-based tool to perform user-guided segmentation while taking both local and global perspectives into account. Thus CNAVIZ enables the user to acquire the advantages of both approaches while overcoming their respective limitations. On simulated data, we demonstrated that CNAVIZ enables the user to produce more accurate segmentation solutions regardless of whether it is run in *de novo* mode or used to refine local or global segmentations. On real data, we demonstrated an example of how CNA analyses are afforded tangible downstream improvements when we perform manual editing in the CNAVIZ user interface. CNAVIZ is open source and is available at: https://github.com/elkebir-group/cnaviz. The most recent version of CNAVIZ is deployed at: https://elkebir-group.github.io/cnaviz.

There are several avenues for future research. First, while the 'Cluster Analytics' tab provides static feedback on the current segmentation, we envision the tool could provide real-time suggestions to further improve segmentation. Second, CNAs are often recurrent across patients with the same tumor type. Presently the tool operates on samples from one tumor at a time. In the future, we may consider generating suggestions based on segmented data from tumors in the same cohort. This will help further automate the process of generating and improving segmentation. Third, while this manuscript focused on applications of CNAVIZ to bulk DNA sequencing data, CNAVIZ is also applicable to single cell DNA sequencing data. We refer the reader to Fig D in S1 Text for an example. Compared to bulk DNA sequencing data, the lower coverage in single-cell data results in fewer bins that span larger genomic regions (e.g. the single-cell data illustrated in Fig D in S1 Text has 5 MB bins as compared to 50 KB bins in bulk whole genome sequencing data). However, the main challenge is that the number

of samples in single cell data, which can be as large as 1, 000 cells [42, 43], far exceeds CNA-Viz's capacity for effective visualization and comparisons across samples. Thus, although CNAViz can be used to visualize single-cell DNA sequencing data, it will likely require some changes to improve the analysis across samples. Moreover, we envision the interface to aid the user to detect doublets [44] as well as determine cell-specific scaling factors used in downstream copy-number calling [22]. We leave this to future work. Finally, we propose an opt-in way for users to contribute segmentation solutions akin to crowd-sourcing efforts like FoldIt, enabling future developments of automated segmentation algorithms that incorporate successful strategies employed by expert users [45].

## Supporting information

**S1 Text. Supplementary materials.**
(PDF)

## Acknowledgments

We thank Brian Arnold for providing us with scripts to run HATCHet in a modular fashion.

## Author Contributions

**Conceptualization:** Gillian Chu, Mohammed El-Kebir.

**Data curation:** Gillian Chu.

**Formal analysis:** Gillian Chu, Silas Hsu, Simone Zaccaria, Mohammed El-Kebir.

**Funding acquisition:** Simone Zaccaria, Mohammed El-Kebir.

**Investigation:** Gillian Chu, Simone Zaccaria, Mohammed El-Kebir.

**Methodology:** Zubair Lalani, Gillian Chu, Silas Hsu, Shaw Kagawa, Michael Xiang.

**Project administration:** Mohammed El-Kebir.

**Software:** Zubair Lalani, Gillian Chu.

**Supervision:** Simone Zaccaria, Mohammed El-Kebir.

**Validation:** Gillian Chu, Simone Zaccaria.

**Visualization:** Zubair Lalani.

**Writing – original draft:** Zubair Lalani, Gillian Chu, Silas Hsu, Shaw Kagawa, Michael Xiang, Simone Zaccaria, Mohammed El-Kebir.

**Writing – review & editing:** Gillian Chu, Simone Zaccaria, Mohammed El-Kebir.

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
