## [Decision Letter · Decision Letter 0]

18 Aug 2022

Dear Dr. El-Kebir,

Thank you very much for submitting your manuscript "CNAViz: An interactive webtool for user-guided segmentation of tumor DNA sequencing data" for consideration at PLOS Computational Biology. As with all papers reviewed by the journal, your manuscript was reviewed by members of the editorial board and by several independent reviewers. The reviewers appreciated the attention to an important topic. Based on the reviews, we are likely to accept this manuscript for publication, providing that you modify the manuscript according to the review recommendations.

In addition the reviewers suggested that  the paper would fit better the Software Section of PloS CB.

I agree. Please review the instructions to authors and reformat if needed. 

Sincerely,

Teresa M. Przytycka

Academic Editor

PLOS Computational Biology

Jason Papin

Editor-in-Chief

PLOS Computational Biology

[LINK]

Reviewer's Responses to Questions

**Comments to the Authors:**

Reviewer #1: I reviewed this paper for RECOMB so I am reviewing it in the spirit of a revision even if it is technically a new submission to PLOS CB. Overall, I retain some of my original concerns about the appropriate criteria for evaluating a visualization tool in this space, but I believe the authors do show the software to be useful to the community. They have been very responsive to the critiques of the conference version and I believe have addressed the criticisms pretty much as well as would be possible without making it a substantially different contribution. They have added a lot of new features to the software and new demonstrations and clarifications, as well as improved documentation and instructional videos. The nature of the work still raises some concerns about whether it is computational enough for a computational biology forum as well as whether it is adequately assessed by the standards of a visualization paper. The authors might reconsider in revision whether any more can be done to address those points. But I believe the work is meritorious despite these concerns and the authors make a reasonable argument not to do more on those issues. I do not have any new concerns to raise at this point and am satisfied that the contribution is sound.

Reviewer #2: General:

CNAViz is a webtool that allows the user to visualize and manually correct the segmented copy number sequencing data for whole genome DNA bulk sequencing. The user can also perform de novo manual segmentation. The graphical interface makes the tool accessible even for unexperienced users thanks to the user manual that not only explains how to use CNAViz, but also how to prepare the data for the analysis. The authors clearly justify why such manual inspection and correction of copy-number segmentation is needed in the process of analyzing CNAs in cancer.

Major:

1. The authors acknowledge the existence of two very different types of DNA sequencing data - bulk and single cell. This is indeed a very important subject and I believe that the authors should state explicitly, preferably in the abstract, whether CNAViz in the current form is applicable to both, or only one of those two types of DNAseq data. Additionally, in the paragraph with the overview of existing CN calling algorithms, the methods for bulk sequencing should be separated from scDNAseq methods. The single cell data suitable for CN calling is significantly different from bulk- for example the coverage for whole genome scDNAseq is incomparably lower (usually less than 0.05X) and RDR or BAF info is not easily obtainable. It is my understanding that CNAViz is right now a tool for bulk DNAseq data and this should be repeated in this paragraph together with a detailed overview of relevant existing methods for bulk. scDNAseq should of course be acknowledged, but in a separate sentence. If the authors want to dwell into existing methods for CN calling in scDNAseq data, they should also explain that it presents different challenges and the possibility of addressing them by CNAViz. This is touched upon in the discussion, which I believe is the correct place, but should be expanded on since the scalability is not the only and the biggest challenge for inferring CNs in scDNAseq.

If I am mistaken and the current version of CNAViz is applicable to scDNAseq, at least one example of such analysis should be given.

Minor:

1. The switching back to “Zoom” Mode does not work after I have used the “Select” mode. Sometimes I need to clear the browsing data and cookies and then refresh the page to be able to use the “Zoom” again.

2. The name of “Erase” mode is not accurate as it may suggest that is removes clustering. I would simply change to “Deselect”.

3. The abstract would benefit from removing of repetition of “important” in the following sentence: “Importantly, inaccurate segmentation will lead to inaccurate identification of important CNAs.

4. In the abstract, is: “To improve copy-number segmentation and their control”, should be: „To improve copy-number segmentation and its control”.

5. Lines 8 and 9, is: Not only is the identification of CNAs key to understanding cancer evolution”, should be „Not only is the identification of CNAs a key to understanding cancer evolution”.

6. Lines 24 and 25, is: „Thus, analyzing variations of RDR and BAF values across bins allow the identification”, should be: „Thus, analyzing variations of RDR and BAF values across bins allows the identification”.

7. Line 258, is: “users can choose a cluster I”, should be: „users can choose a cluster i”.

8. Lines 262 to 264 – I would explicitly name the buttons on the sidebar and inform the user that „Undo cluster” also works for reversing the quite powerful “Clear clustering” button. Also “Clear clustering” should work locally i.e. clear clustering for only the selected area.

9. Line 294, is: “and restricted ourselves”, should be: “and restrict ourselves”.

10. Line 296, is: “genomic region that the driver gene spans is highlighted.”, should be: „ genomic region of the driver gene is highlighted.”

11. Line 297, should be Figure 2j.

12. Some problems with how the axis text is double printed in the purity/ploidy mode

13. Line 310, is “Any pair”, should be: “All pairs”.

14. Line 313, is “which, should be “that”.

15. Line 393, is “from”, should be “From”.

**Have the authors made all data and (if applicable) computational code underlying the findings in their manuscript fully available?**

Reviewer #1: Yes

Reviewer #2: Yes

PLOS authors have the option to publish the peer review history of their article (what does this mean?). If published, this will include your full peer review and any attached files.

Reviewer #1: No

Reviewer #2: No

Figure Files:

Data Requirements:

Reproducibility:

References:

---

## [Decision Letter · Decision Letter 1]

29 Sep 2022

Dear Dr. El-Kebir,

We are pleased to inform you that your manuscript 'CNAViz: An interactive webtool for user-guided segmentation of tumor DNA sequencing data' has been provisionally accepted for publication in PLOS Computational Biology.

Best regards,

Teresa M. Przytycka

Academic Editor

PLOS Computational Biology

Jason Papin

Editor-in-Chief

PLOS Computational Biology

Reviewer's Responses to Questions

**Comments to the Authors:**

Reviewer #1: My criticisms were all discretionary so I am fine with the authors declining to make some changes. I think the revisions they have made improve the paper and have nothing more to add.

Reviewer #2: I am satisfied with the improvements implemented in the manuscript and the CNAViz itself.

Just one minor comment, since the single cell field is currently in bloom and many new methods are published, I would change the sentence: "Additionally, the methods HMMcopy and Ginkgo have been developed for single cell DNA sequencing data." to "Additionally, methods such as HMMcopy and Ginkgo have been developed for single cell DNA sequencing data."

**Have the authors made all data and (if applicable) computational code underlying the findings in their manuscript fully available?**

Reviewer #1: Yes

Reviewer #2: Yes

PLOS authors have the option to publish the peer review history of their article (what does this mean?). If published, this will include your full peer review and any attached files.

Reviewer #1: No

Reviewer #2: No

---

## [Editor Report · Acceptance letter]

7 Oct 2022

PCOMPBIOL-D-22-01002R1 

CNAViz: An interactive webtool for user-guided segmentation of tumor DNA sequencing data

Dear Dr El-Kebir,

I am pleased to inform you that your manuscript has been formally accepted for publication in PLOS Computational Biology. Your manuscript is now with our production department and you will be notified of the publication date in due course.

With kind regards,

Agnes Pap
